# Field Investigation and Economic Benefit of a Novel Method of Silver Application to Ceramic Water Filters for Point-Of-Use Water Treatment in Low-Income Settings

Nkosinobubelo Ndebele [1], Joshua N. Edokpayi [1,*], John O. Odiyo [1] and James A. Smith [2]

1    Department of Hydrology and Water Resources, University of Venda, Private Bag X5050,
     Thohoyandou 0950, South Africa; nkosinobubelondebele@live.com (N.N.); John.Odiyo@univen.ac.za (J.O.O.)
2    Department of Engineering Systems and Environment, University of Virginia,
     Charlottesville, VA 22904-4742, USA; jas9e@virginia.edu
*    Correspondence: Joshua.Edokpayi@univen.ac.za; Tel.: +27-788-162-538

**Abstract:** In this study, we report on field testing of ceramic water filters (CWFs) fabricated using a new method of silver application (using silver nitrate as a raw material) compared to conventionally manufactured CWFs (fabricated with silver nanoparticles). Both types of filters were manufactured at the PureMadi ceramic filter production facility in Dertig, South Africa. Thirty households received filters fabricated with silver nitrate ($AgNO_3$), and ten of those households were given an extra filter fabricated with silver nanoparticles. Filter performance was quantified by measurement of total coliform and *Escherichia coli* (*E. coli*) removal and silver residual concentration in the effluent. Silver-nitrate CWFs had removal efficiencies for total coliforms and *E. coli* of 95% and 99%, respectively. A comparison of the performance of silver-nitrate and silver-nanoparticle filters showed that the different filters had similar levels of total coliform and *E. coli* removal, although the silver nitrate filters produced the highest average removal of 97% while silver nanoparticles filters recorded an average removal of 85%. Average effluent silver levels were below 10 ppb for the silver-nitrate and silver-nanoparticle filters, which was significantly below the Environmental Protection Agencies of the United States (EPA) and World Health Organization (WHO) secondary guidelines of 100 ppb. Silver-nitrate filters resulted in the lowest effluent silver concentrations, which could potentially increase the effective life span of the filter. A cost analysis shows that it is more economical to produce CWFs using silver nitrate due to a reduction in raw-material costs and reduced labor costs for production. Furthermore, the production of silver-nitrate filters reduces inhalation exposure of silver by workers. The results obtained from this study will be applied to improve the ceramic filtration technology as a point-of-use (POU) water treatment device and hence reduce health problems associated with microbial contamination of water stored at the household level.

**Keywords:** ceramic water filter (CWF); point-of-use (POU) water treatment technologies; silver nitrate; silver nanoparticles; waterborne diseases; public health

## 1. Introduction

Access to clean, safe, and adequate amounts of water is a fundamental human need and, therefore, a basic human right. Microbes such as viruses, bacteria and protozoa are easily transported through drinking water. Ingestion of such pathogens in water leads to the greatest water-related health risks and is a major cause of waterborne diseases [1,2]. In most developing countries, there is an erratic supply of treated water to rural communities. Hence rural dwellers often store treated water (if available) for a long period of time (which encourages recontamination) before use, and most of them resort to the use of untreated water from rivers, wells and boreholes, which are often plagued with the presence of pathogens [3]. This problem is exacerbated by poor sanitation and hygienic practices. Chemical contaminants (like fluoride and arsenic) in water are often geogenic in nature and

restricted to a particular environment [4]. Previous studies have shown that the microbial quality of drinking water is the most common water-related problem in the world [5,6].

Developing countries report the highest number of deaths related to waterborne diseases, particularly in rural areas [1,2]. It is increasingly difficult for local governmental bodies to set up centralized water treatment plants and provide potable water via a piped distribution system due to inadequate funds and the lack of basic infrastructure in rural areas [7]. As developing countries continue to encounter water quality challenges, it is imperative to focus on the provision of safe drinking water rather than focusing on the provision of high-quality large volumes of water for all uses. Due to the low economic status in developing countries, options of low-tech water treatment should possess at least the following characteristics:

- A technology that is produced using locally available materials and labor;
- A technology that is cost-effective and easy to manufacture;
- A technology that is easy to operate and socially acceptable so that users are willing to maintain it to enable it to last for its maximum possible lifetime [8].

Thus, more practical options for low-income countries could include using proven point-of-use (POU) water treatment methods. In rural areas of South Africa, POU water treatment technologies are being widely promoted by the government and other organizations as an appropriate intervention for reducing the burden of waterborne diseases. Ceramic water filters (CWFs) are examples of POU water treatment technologies that are being used by rural communities in South Africa. CWFs are usually produced by firing a mixture of locally available materials, which include suitable clays, burn-out materials (e.g., rice husks and sawdust) and water. Silver nanoparticles (AgNPs) are the major disinfectants that are usually added to CWF to aid microbial inactivation and to prevent the recontamination of treated water and the growth of biofilms on the surface of the filters [9,10].

Various methods have been reported on the mode of silver nanoparticles (AgNPs) in addition to CWF. This includes the painting of colloidal silver on the surface of the filter, dipping of the filters into AgNPs solution and the adding of AgNPs solution as part of the material mixture to make the filter before firing [11].

With the advancement of material development, silver nanoparticles can be easily applied to solid materials for the inactivation of microorganisms in contaminated water [12]. Studies have proven that nanoparticles possess excellent antibacterial characteristics [13,14]. Silver nanoparticles have therefore been widely used to aid drinking water purification due to their broad-spectrum of bactericidal activities [15–17].

Regardless of the success that has been recorded in attempts aimed at the provision of potable water using AgNPs CWFs, research is still ongoing to improve on previous concepts and discover more cost-effective methods that could be applicable, particularly in marginalized areas with low economic status.

PureMadi is one organization that runs a filter manufacturing facility in rural Dertig, North West Province of South Africa and Mukondeni, Limpopo Province of South Africa. CWFs are manufactured using local labor and materials (clay, sawdust and water). Silver nanoparticles are painted to these CWFs to act as a disinfectant towards pathogens, and this method of silver application is used in most filter-making facilities globally.

However, the use and painting method for AgNPs application has several disadvantages:

- AgNPs are not locally available in South Africa and other developing world markets and therefore are imported by filter production facilities;
- Nanoparticles may be released from the filter, particularly in the early-stages of filter use, which can potentially result in silver concentrations in the treated water that are greater than the World Health Organization (WHO) and Environmental Protection Agencies of the United States (EPA) secondary drinking water guideline value of 100 ppb based on health effects [18,19];
- Application of the aqueous nanoparticle suspension is labor-intensive, requiring facility workers to manually paint the solution on the surfaces of every filter;

- Using nanoparticles during the manufacturing process may also constitute a health risk for workers manufacturing the filters, as some research suggests that inhalation of silver particles may result in genotoxic effects [20,21].

The use of silver nitrate in CWF can reduce the risk of inhalation exposure by workers manufacturing CWFs [22]. Few studies have reported the painting of silver nitrate (AgNO$_3$) solution instead of AgNPs on CWF [9–11,23,24]. The results reported show excellent bacteria inactivation. Ryner et al. [11] and Mittelman et al. [10] have both reported that silver retention in CWF made from AgNO$_3$ is poor and can produce treated water with levels of ionic silver (Ag$^+$) exceeding 100 ppb. This study, however, reports our findings on the use of AgNO$_3$ as a chemical that is mixed with other clay forming substances prior to firing. In addition, this is the first study to present data on use of such a novel method of silver application for a period greater than six months in the field.

The aim of this study is to evaluate the comparative microbiological effectiveness of silver-impregnated CWFs made from AgNO$_3$ and the conventional AgNPs methods in households in the Dertig area of South Africa and to compare the cost incurred in their production.

## 2. Materials and Methods

### 2.1. Regional Description of Study Area

The study area is located in Dertig, Bojanala District, North West Province of South Africa (Figure 1). Dertig is governed by the Moretele Municipality, and its geographical coordinates are 25°16′45″ South, 28°13′21″ East [25]. This area of study is conducive as it is where the PureMadi Dertig Ceramic Filter Facility is located [26]. Dertig has a total population of 2996 and 786 households [27].

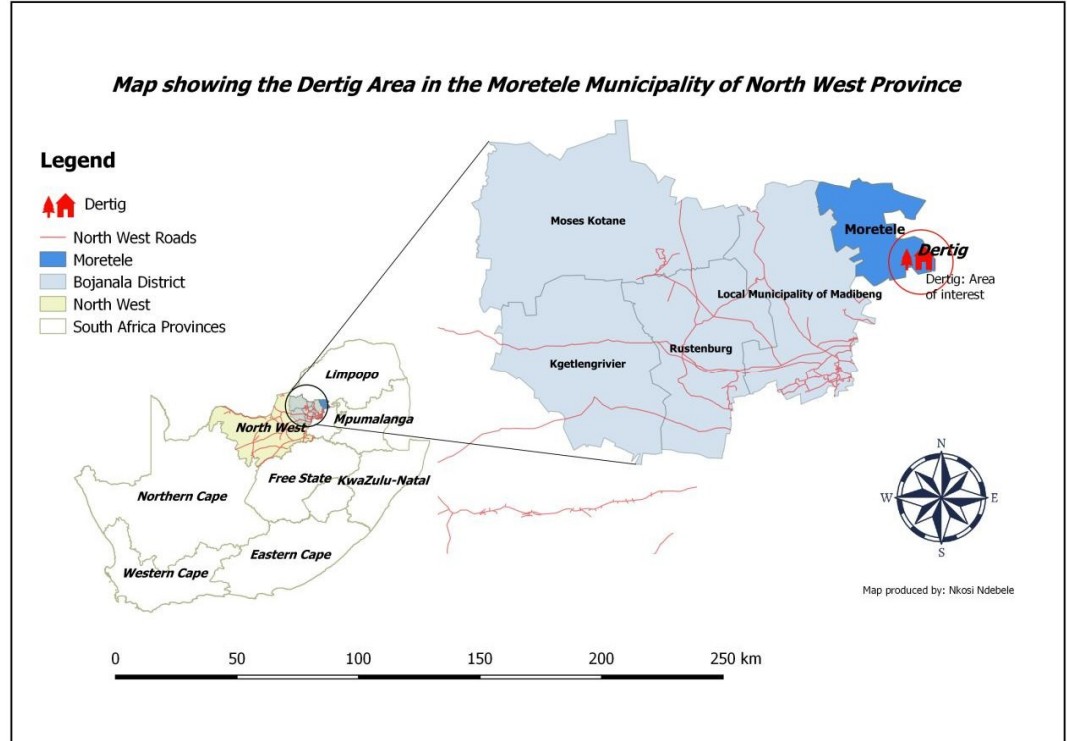

**Figure 1.** Study area: South Africa.

### 2.2. Study Design

A baseline census of the Dertig area was conducted in its two wards (14 and 22) to identify households in which there was at least water interruption that lasted for more than two days and practiced water storage. Thirty households (15 from each ward) were

randomly selected based on the baseline census and willingness to participate in the study. Baseline data were collected from each household, including the assessment of water quality. A sample of drinking water was taken every month during the sampling period, which lasted for 13 months. Ten of the households with five or more residents were given two filters (one with AgNPs and the other with either 1 g or 2 g of Ag prepared from $AgNO_3$) in a receptacle with a spigot. The other 20 households received a CWF prepared from $AgNO_3$. Participating households were trained in the proper use of the CWF by the study team and advised on the importance of properly maintaining the CWF. They were issued with printed filter use instructions during the inception of the study. They were also encouraged to fill the filter every night so that there would be sufficient treated water in the morning for consumption. A follow-up visit was performed monthly for 13 months.

### 2.3. Water Sources in the Study Area

The main water source in the Dertig area is a piped water system from Magalies Water Treatment Works. Due to water rationing, residents in the Dertig area suffer from 3–7 day water interruptions every week. Water rationing has, therefore, forced families to store enough water for usage during times when there is no running water from the taps. Secondary water sources in the Dertig area include groundwater (from boreholes, some with protected hand pumps), rainwater harvesting and water from trucks. Tshwane River water is only used for other domestic purposes except drinking.

### 2.4. Production of CWFs

In this study, three kinds of CWF were used. The first kind of CWFs (10) were purchased from the PureMadi Dertig Ceramic Filter Facility, made from clay (34 kg), sawdust (4 kg) and water (16.5 L). Details of how the filters were produced can be obtained from a previous publication and at the company's website [22,26]. The final step of the filter production involves the application of AgNPs produced from collargol (70% Ag by mass, Laboratorios Argenol S.L, Zaragoza, Spain). The inside and outside of the filter were painted with the solution for a total of 0.4 g of silver in each filter. Twenty-five CWFs were produced, each containing 1 g and 2 g of ionic silver from $AgNO_3$. Clay (50 kg), sawdust (22 kg), grog (a substance made by crushing fired filters that have failed the quality tests) (5 kg), water (20 L) and $AgNO_3$ (analytical grade, 99% pure; Chem Lab Supplies, South Africa) were used. To obtain 1 g and 2 g ionic silver per filter, 39.37 g and 78.74 g of $AgNO_3$ were employed, respectively. The mixing of the various components of the filters, pressing and drying, firing, cooling, quality test and packaging were performed as reported by Jackson et al. [22]. A greater quantity of the $AgNO_3$ -based CWF was produced to compensate for those that may fail one of the quality tests and to replace any broken filter during transportation and handling. Thirty of the filters were used for the field study.

#### 2.4.1. Quality Control

Visual inspections took place before each major step of the production process so that defective filters could be removed from the production line. Formal visual inspections were carried out before surface finishing, loading the kiln, flow rate testing, silver application and packaging. Filters were examined for cracks, warping, inconsistent filter walls, large pieces of burn-out material and consistent surface finish. In fired filters, filters were examined for discoloration, including blackened areas indicating incomplete combustion of the burn-out; warping; cracks; holes or spaces from large pieces of burn-out material; charring; crumbling; and that the base and rim of the filter was at the proper angle to the wall of the filter. The filter rim of fired filters was checked for size and warping by placing a receptacle lid on each filter element. The lid was turned slowly, and it was checked that the filter rim meets the lid evenly. If the lid did not fully cover the filter rim, the filter element was ground using caution not to damage the body of the filter or grind any more material than necessary [28].

### 2.4.2. Health and Safety

Health and safety measures were ensured to reduce any risk associated with filter-making, such as inhalation of dust particles during sieving of the sawdust and throughout the production of the CWFs. Personal protecting equipment such as gloves, goggles and dust masks were provided and used at all times.

### 2.5. Water Sampling

Drinking water samples, regardless of the source, were collected monthly from the 30 households into sterile containers. The physicochemical and microbiological water quality parameters (total coliform and *E. coli*) from the source water were measured monthly for 13 months. Similarly, the filtered water was also collected using the spigot in the receptacle of the bucket containing the CWF into a sterile container, and the levels of total coliform and *E. coli* were also enumerated. This was done monthly for 13 months for the different kinds of filters. Distilled water from the Hydrology and Water Resources laboratory of the University of Venda was used as a control sample for physicochemical and microbiological analyses.

### 2.6. Measurement of Physicochemical Parameters

Physicochemical parameters of source water samples were measured in the field by a YSI Professional Plus meter (YSI Inc., Yellow Springs, OH, USA) for pH and conductivity. The probes and meter were calibrated according to the manufacturer's instructions. Turbidity was measured in the field with an Orbeco-Hellige portable turbidimeter (Orbeco-Hellige, Sarasota, FL, USA). The turbidimeter was calibrated according to the manufacturer's instructions. Measured levels were compared to the South African National water-quality standards.

### 2.7. Microbiological Water Analysis

Analysis of the samples was carried out within 24 h of sample collection. Water samples from ceramic filters made with silver nitrate and silver nanoparticles were evaluated for total coliforms and *E. coli*. Systematic measurement and observation of the two microbial parameters were carried out at the PureMadi Ceramic Filter Facility. The membrane filtration technique was used to detect the presence of microorganisms in water. As a disinfection measure, manifold sample cups were placed in a boiling water bath set to 100 °C for 15 min. Filter paper disks of 47 mm diameter and 0.45 micropore size ($4.5 \times 10^{-7}$ m pore size) (EMD Millipore, Billerica, MA, USA) were placed on the surface of the manifold with forceps with the following aseptic techniques. 100 mL samples were passed through the filter paper. The filter papers were transferred to a sterile Petri dish that had an absorbent pad of selective growth media solution (m-ColiBlue24, EMD Millipore, Billerica, MA, USA). The samples were incubated at 35 °C for 23 to 25 h. Total coliform and *E. coli* colonies were counted and reported as colony-forming units per 100 mL (CFU/100 mL) of water sample [3]. Distilled water was used for the control experiment.

### 2.8. Measurement of Silver Levels in Effluent

Graphite furnace atomic absorption spectrophotometry (AA2100; PerkinElmer, Waltham, MA, USA) (GFAA) was used for the quantification of silver in the filtered water. Ten millimeter samples from the filtered water were collected from each participating household monthly and stored in a refrigerator. The samples were analyzed at the Department of Engineering Systems and Environment at the University of Virginia. Before analysis, the samples were prepared with nitric acid (1%) to reduce the chelation of ions [29]. A total of 42 samples (14 samples each of the three filter types) were randomly selected and transported for analysis using the GFAA.

### 2.9. Economics of the Process

An analysis of the economics involved in the production was carried out to assess the economic benefit of using silver nitrate instead of silver nanoparticles in filter making. Since the labor cost, cost of clay and other production materials is the same for all kinds of filters, only the cost of silver nitrate and AgNPs were used to compute this. In addition, the shipping cost was included in the cost of the AgNPs. The unit cost analysis was employed to ascertain the cost involved in producing 1000 units of each kind of filter.

### 2.10. Ethical Consideration

First, ethical authorization was obtained from the ethics committee of the University of Venda. Consent to carry out the study was then requested from PureMadi, the implementers of the Ceramic water filter technology in Dertig, South Africa.

Prior to sample collection, permission was requested from the Moretele Municipality and Dertig community leaders. Consent was then requested from the volunteering households where they were also informed about the purpose of the research, details of their participation, how collected data will be used as well as the benefits of the study.

The research involved using different laboratory chemicals in microbial water analysis. Hence there was safe handling and safe disposal of cultures, reagents, and materials and while operating sterilization equipment to protect the health of the individuals working at the filter facility as well as safeguarding the environment at large.

### 2.11. Statistical Analysis

Excel version 26 was used to analyze the samples statistically using One-way Analysis of Variance (ANOVA). Delta graph was used for some of the plots.

## 3. Results and Discussion

### 3.1. Socio-Demographic Characteristics of Enrolled Households

Thirty households enrolled in the study and responded to a survey data questionnaire. The highest range of people per household was between 4 to 6 (n = 19, 63%) as shown in Table 1.

**Table 1.** Number of people per household.

| Range of Number of People per Household | Number of Households | Percentage |
|:---:|:---:|:---:|
| 1–3 | 7 | 23% |
| 4–6 | 19 | 63% |
| 7–9 | 4 | 14% |

Adult women are most often responsible for water management (n = 23, 77%) at home, while adult men are least often responsible for managing water (n = 7, 23%). One hundred percent of households have their primary water source piped to their yards, with all households reporting the origins of their water to be a municipal treated source (n = 30). One hundred percent of the households suffer 3–7-days water interruptions weekly. Because of prevailing water supply interruptions in the Dertig area, most enrolled households store their drinking water in plastic buckets (n = 24, 81%), while a few households store their water in plastic bottles (n = 6, 19%).

Most households (n = 24, 80%) fill their storage containers directly from the tap, while only a fifth of the households (n = 6, 20%) use hosepipes to fill their storage containers. Collection of water from storage containers is through cups with handle (n = 30, 100%). All households cover their stored water with a lid. When the stored water is used up, most households (n = 17, 57%) have their secondary water source from tanker trucks (delivered at a central community point every 2 days), while some (n = 7, 23%) households use rain harvested water from their JoJo tanks. Only a few households (n = 5, 17%) have their secondary water source from nearby boreholes.

Forty-seven percent of the respondents describe their drinking water quality as average as it is sometimes cloudy and smells bad. Some households describe their drinking water as poor (n = 12, 39%), while a few households describe their drinking water as very good quality (n = 4, 14%) (Figure 2*)*.

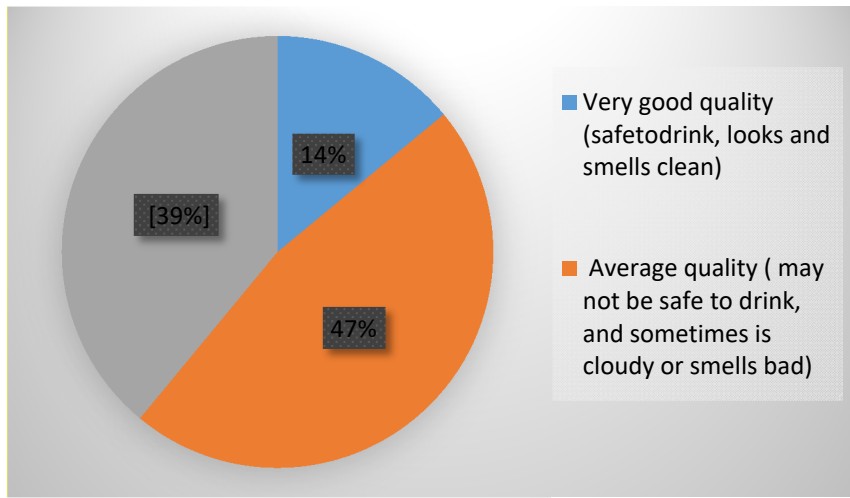

**Figure 2.** Baseline perception of water quality by households.

*3.2. Characterization of Raw Water*

Raw water in this context refers to the water obtained from the homes of the respondents, which could be municipal stored water and water from rainwater harvesting and boreholes.

3.2.1. Physicochemical Parameters of Raw Water

Physicochemical tests carried out on all raw water samples at the beginning of the study included conductivity, turbidity, total dissolved solids, color and pH. The test results showed that raw water from households in the Dertig area had conductivity, total dissolved solids and pH within the recommended South Africa National Standards (SANS) for drinking water quality limits (Table 2) [30]. However, color and turbidity levels were above the recommended limit set by SANS for drinking water. This result corroborates the findings from the baseline data, where most of the participants agree that their water is sometimes cloudy (n = 14, 47%).

**Table 2.** Results showing average values of physicochemical determinants of raw water (n = 780) used in filter performance tests with reference to SANS for drinking water quality.

| Water Quality Parameter | Average Raw Water Concentration | St. Dev for Raw Water Concentration | Risk | SANS Drinking Water Standards [30] |
|---|---|---|---|---|
| Conductivity | 120 mS/m | 21.89 | Esthetic | ≤170 |
| Total dissolved solids | 1150 mg/L | 67.72 | Esthetic | ≤1200 |
| Color | 16 mg/L as Pt-Co | 1.145 | Esthetic | ≤15 |
| Turbidity | 2 NTU | 0.695 | Operational and esthetic | ≤1 |
| pH | 8 | 1.083 | Operational | ≥5 and ≤9.7 |

In this study, high color levels in raw water could be due to the frequent water interruptions in the Dertig area. A policy position statement issued by The Chartered Institution of Water and Environmental Management (CIWEM) [31] emphasizes that water color changes in areas where water interruptions are prevalent are commonly associated with

effects occurring within the water distribution network (for example, the reintroduction of flow following an interruption) rather than problems at the water treatment plants.

On the other hand, WHO [32] explains that at times turbidity can be an indication of the presence of microbes and, therefore, an indicator of contamination in the water supply system from source to POU. The findings of a study by Ogutu et al. [33] indicate that the poor execution of water treatment steps (such as coagulation, filtration and chemical disinfection) at the treatment facility may lead to high turbidity levels. In addition, the effectiveness of water distribution system management may highly influence turbidity levels. They further explain that leaks within the distribution system can be another factor for a sudden increase in turbidity recorded in household water that is obtained directly from the distribution mains. This is because a difference in atmospheric pressure inside the pipes can cause dirt and particles such as sand to be sucked into the distribution system during times when the water velocity is low.

POU water treatment technologies such as CWFs may be affected by high turbidity levels in the source of water, and this may strongly affect their lifespan and effectiveness in water purification [32]. One advantage of CWF is that it provides a barrier system for the removal of suspended solids and other turbidity causing substances from drinking water as well as microorganisms by using a combination of chemical and physical processes [34].

### 3.2.2. Microbiological Parameters of Raw Water

A summary of the mean of total coliform and *E. coli* of the raw water (before filtration) from different water sources in the Dertig area as a function of time is presented in Figures 3 and 4. Some household raw water samples tested positive for both total coliform and *E. coli*. The majority of the households, however, tested negative for *E. coli*. The average levels of *E. coli* recorded for some sampling months was high due to the value recorded in a few of the households. Generally, low levels of *E. coli* were determined in the household source water. The presence of both indicator organisms in some of the drinking water implies that the water is not safe for drinking and could possibly put the consumers at risk of waterborne diseases hence the need for a point-of-use water treatment system.

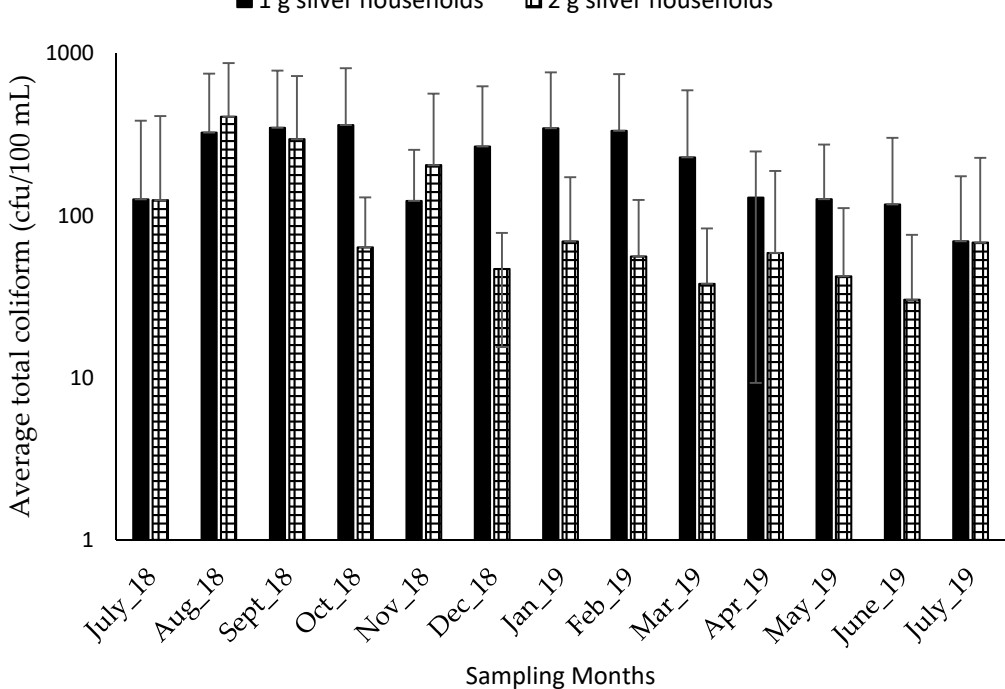

**Figure 3.** Average values of total coliform of raw water (n = 390).

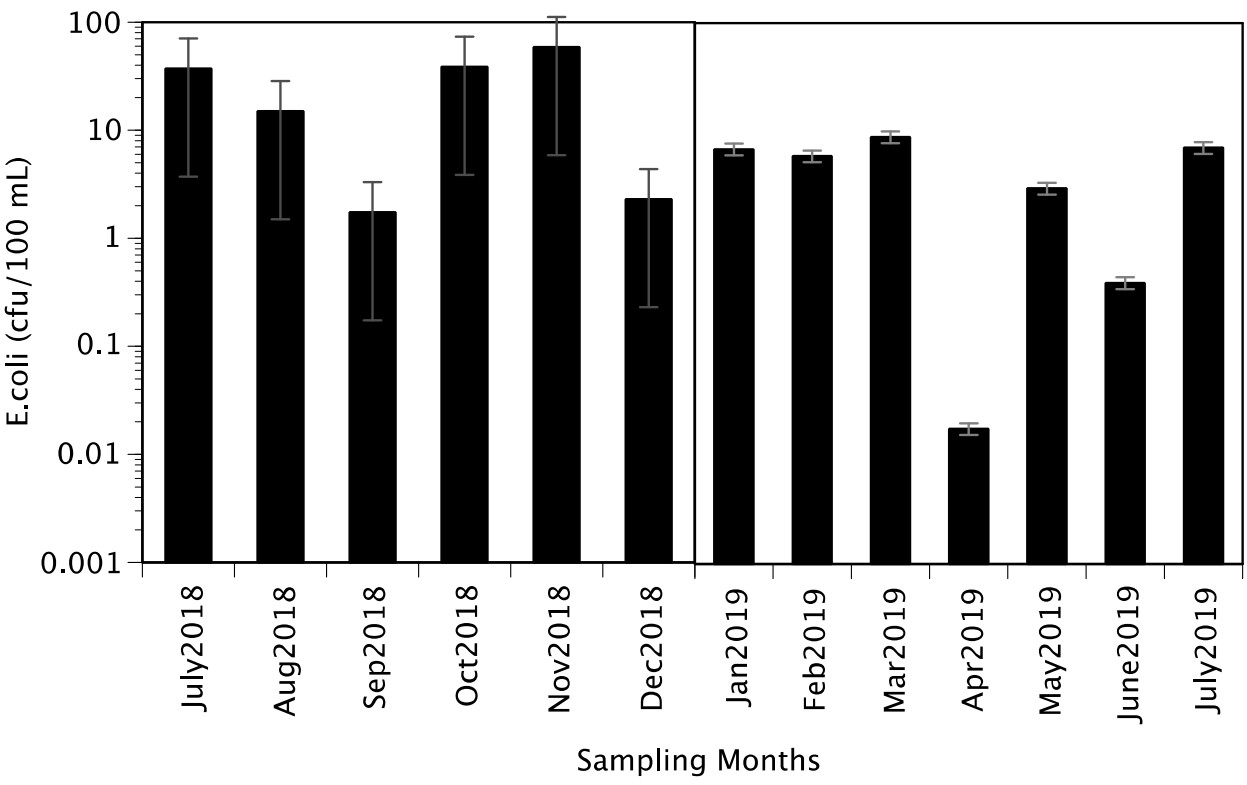

**Figure 4.** Average values of *E. coli* of raw water (n = 390).

At the beginning of the study, 100% of households reported their primary source of drinking water is municipally treated water. However, during the period of rationing, they either use their stored water or get water from secondary sources, which are often untreated. Therefore, the presence of indicator organisms in some of the household water could be due to inadequate municipal treatment of water, the poor microbiological water quality of the secondary source of water and possible recontamination of treated water during storage and use.

One-way ANOVA was used to ascertain if the microbial level of the raw water from the households that receive CWF of 1 g and 2 g were statistically different. The results obtained showed that the households that receive both kinds of filters vary significantly with the levels of total coliforms ($p < 0.05$), but the levels of *E. coli* in the household samples did not vary significantly ($p > 0.05$).

### 3.3. Microbiological Performance of CWFs made with Silver Nitrate

3.3.1. Removal Efficiency of Total Coliform

Results show that CWFs made with silver nitrate recorded a high removal efficiency for total coliforms (95%). This implies that incorporating silver nitrate before firing the filters is effective in inactivating total coliform in contaminated water. A previous study carried out by Mwabi et al. [35] proved that CWFs impregnated with silver nanoparticles were efficient in producing water that is microbiologically safe to drink, regardless of the type of water source. These silver nanoparticles impregnated CWFs had better performance than some other POU water treatment devices such as bucket filters, bio-sand filters and ceramic candle filters [35]. The high-performance of CWFs made with silver nitrate could be attributed to the antibacterial properties of the ionic silver in $AgNO_3$ that is mixed with clay, sawdust, grog and water during the manufacturing process.

Figure 5 gives a summary of the distribution of total coliform in both raw and filtered water. The total coliform median value for raw water decreases from 160 CFU/100 mL to 2 CFU/100 mL in the treated water after filtration. Ceramic water filters are therefore effective in the reduction of total coliforms in drinking water. Although a 0 CFU/100 mL is

recommended by SANS drinking water standards, the presence of total coliform does not indicate the health risk of the water to the consumer [30]. Safe storage and the practice of good hygiene will aid in achieving the required value.

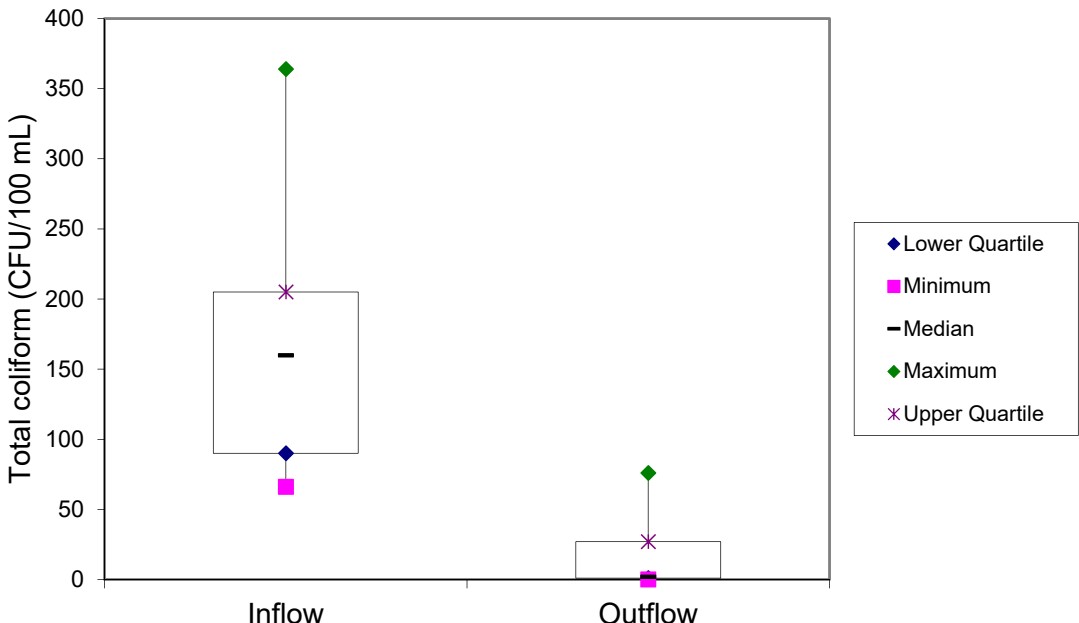

**Figure 5.** Box and whisker plot highlighting differences of inflow (n = 390) and outflow (n = 390) total coliform count for the silver nitrate filters.

CWFs made with silver nitrate were effective in reducing total coliform throughout the 13-month period, and their long disinfection capacity could be due to the release of the silver ion into the water. A similar study conducted by Nangmenyi et al. [36] has documented the effect of silver in fiberglass water treatment. Silver is known to have antibacterial properties that inactivate bacteria by disrupting the disulfide bond formation of proteins in the cell membrane or by inhibiting DNA synthesis [37,38]. It is thus possible that silver was a major contributing factor towards the removal of bacteria in drinking water consumed by Dertig families who participated in the study.

### 3.3.2. Removal Efficiency of *E. coli*

Water samples from the 30 participating households were collected and tested for *E. coli* prior to and after treatment by CWFs made with silver nitrate. The removal efficiency of *E. coli* proved to be very high, at 99%. A similar study assessing the effect of activated metallic silver on water quality in a laboratory setup was reported by Meierhofer et al. [39], where fecally contaminated tap water containing more than 1000 CFU/100 mL of *E. coli* was used. Results showed that *E. coli* was completely inactivated in batches containing metallic silver after about 12 h, and *E. coli* coliforms were not inactivated in the control configurations, which contained no silver. Figure 6 gives a summary of the distribution of *E. coli* in both raw and filtered water. The *E. coli* median value of raw water was 2 CFU/100 mL, thus proving that the household drinking water did not comply with WHO and SANS drinking water standards. There is, therefore need for POU water treatment before consumption. There was no *E. coli* count in the ceramic filtered water, and this proves that ceramic water filters made using silver nitrate produce water that complies with WHO and SANS drinking water standards and is safe for human consumption.

Proper water handling, hygiene practices and safe storage are essential in the provision of good water quality as they prevent recontamination and offer long-time inactivation of bacteria by silver during storage. Whenever the researchers visited households for sample collection, water was continuously stored in the ceramic filter receptacles. A possible

factor contributing to the high removal efficiency of *E. coli* by the CWFs made with silver nitrate could also be linked to the contact time with silver during storage [38]. Van der Laan et al. [40] carried out a study to determine the role of silver during filtration and subsequent storage. Results showed that storage time in the receptacle contributed to the inactivation of *E. coli* by silver, to a great extent. The study concluded that water storage time after filtration determined *E. coli* inactivation efficacies rather than CWF characteristics such as sawdust and clay.

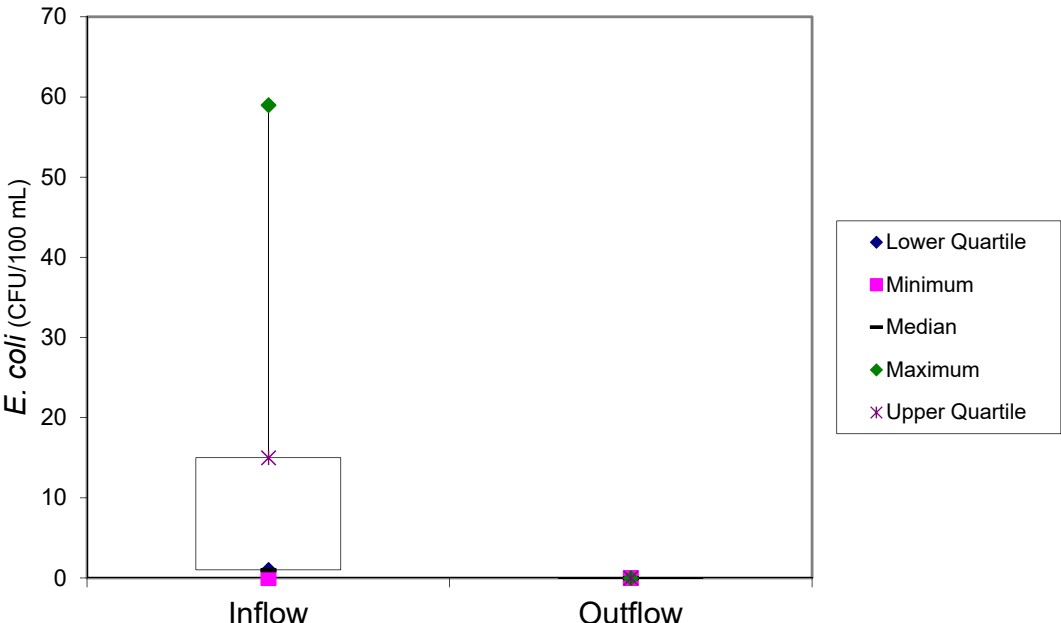

**Figure 6.** Box and whisker plot highlighting differences in inflow (n = 390) and outflow (n = 390) of *E. coli* count. for the silver nitrate filters.

*3.4. Comparison of Microbiological Quality of Silver Nitrate and Silver Nanoparticles CWFs*

Out of 30 households participating in the study, 15 households had CWFs with 1 g of silver nitrate added during the manufacturing process, while 15 households had CWFs with 2 g of silver nitrate added during the manufacturing process. Ten of those 30 households were randomly selected and given an extra filter with painted AgNPs. A comparison of the microbiological quality of the three types of CWFs within a 13-month period was carried out.

The effectiveness of both filters in improving microbial water quality could be because the filters were still new and had only been used for 13 months (they have a life span of 3 years). New filters with fresh silver coating have been found to be very effective in water purification. A removal efficiency of 99–100% for *E. coli* was established in new filters [37,41]. They further explain that, over time, as the amount of water to be treated increases, silver leaches out of the CWF into the water reducing the efficiency of the filter as a POU water treatment device.

Figure 7 highlights that both methods of silver application (i.e., incorporating silver nitrate before the firing stage and painting-on silver nanoparticles after the firing stage) are effective in total coliform and *E. coli* removal. Silver ions are highly effective in the disinfection of a wide range of waterborne microorganisms [42].

A calculation of the percentage total coliform and *E. coli* removal by 1 g, 2 g and silver nanoparticles filters over 13 months period was carried out, and results show that the different filters result in similar levels of total coliform and *E. coli* removal (Table 3). It found that silver nanoparticles filters had a slightly lower removal efficiency of total coliform (72%) compared to 1 g and 2 g silver nitrate filters.

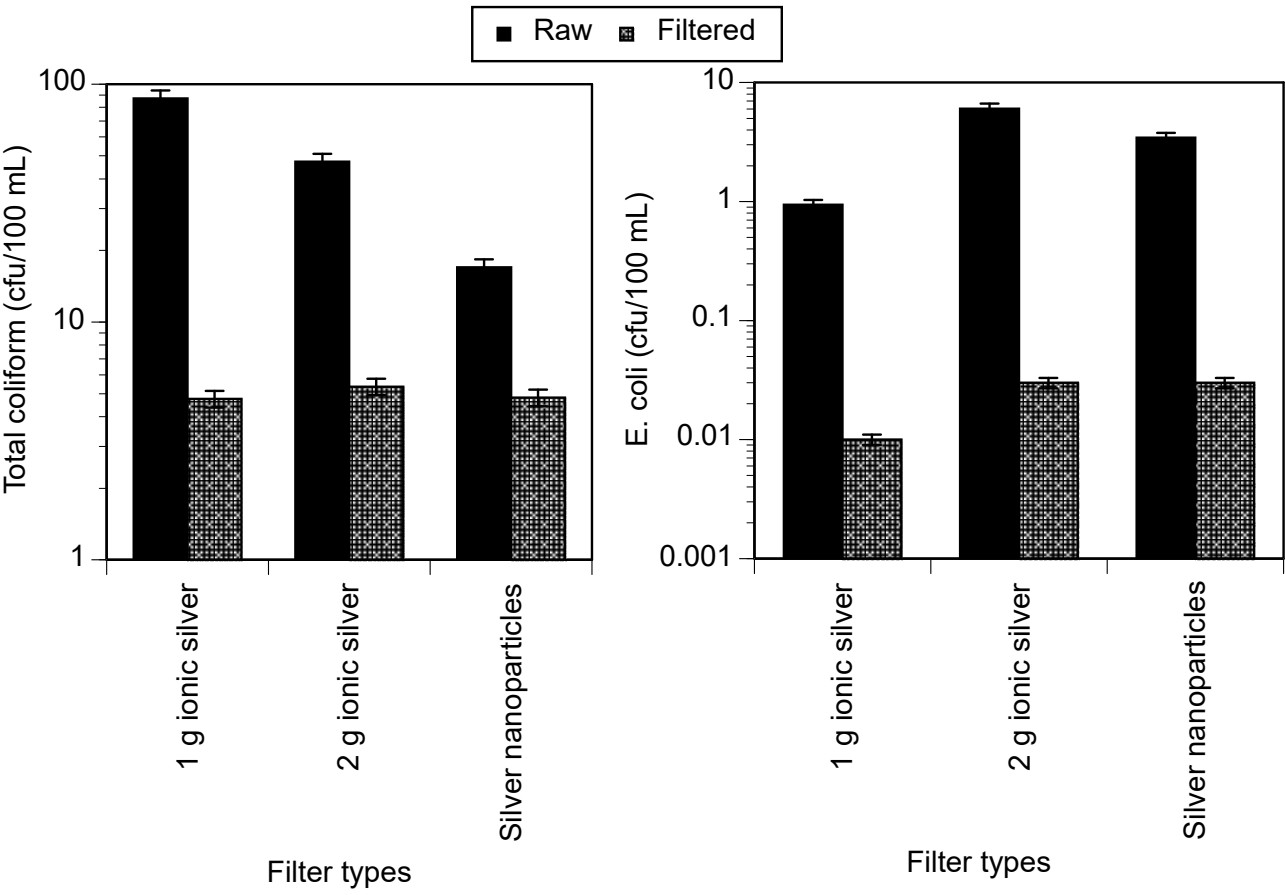

**Figure 7.** Total coliform (**left**) and *E. coli* (**right**) inactivation using the various filter types (n = 1040).

**Table 3.** Percentage coliform removal for total coliform and *E. coli* by 3 filter types.

| Filter Type | Total Coliform Removal | *E. coli* Removal |
|---|---|---|
| 1 g silver nitrate | 96% | 99% |
| 2 g silver nitrate | 89% | 100% |
| AgNPs | 72% | 99% |

The results are in line with Jackson et al. [22], who carried out laboratory studies which recorded that CWFs made using silver nitrate recorded a slightly higher total coliform and *E. coli* removal (log reductions of 4.06 and 4.11) relative to AgNPs CWFs (log reductions of 3.85 and 3.92) although this inactivation efficiencies were not statistically significant ($p > 0.05$).

Therefore, it can be concluded that CWFs made with both 1 g and 2 g silver nitrate and silver nanoparticles had comparable efficiency for bacteria inactivation. It is also noted that there is no significant improvement in performance for filters made with 2 g silver nitrate relative to filters made with 1 g silver nitrate. The performance of the three kinds of filters was also tested using One-way ANOVA, and the bacteria removal efficiency of both total coliform and *E. coli* did not vary significantly ($p > 0.05$), implying that the filters perform comparably though the 1 g Ag CWF recorded a marginal higher bacteria inactivation.

### 3.5. Silver Levels in Effluent

This study has proved that applying both silver nitrate and AgNPs to CWFs improves microbiological efficacy in household water treatment. The application of silver to CWFs also prevents stored water from recontamination. Lyon-Marion et al. [43] assert that if silver added to the CWF during production is above the recommended standard, silver

release in CWF may lead to undesirable health effects. It is therefore very vital for filter manufacturing facilities to add the right amount of silver in order to achieve the goal of POU water treatment and, at the same time, not exceed the maximum recommended silver levels of drinking water.

Average effluent silver levels in this study were $0.07 \pm 0.04$ µg/L (1 g Ag⁺ CWF), $0.6 \pm 1.10$ µg/L (2 g Ag⁺ CWF) and $0.8 \pm 1.0$ µg/L (AgNPs CWF), respectively (below the EPA and WHO guidelines of 100 ppb).

It was observed that silver levels in all filters decrease throughout the study. A possible explanation for this could be the frequency of use of CWFs by households. A study on silver nitrate filters by Kendarto et al. [24] recorded that the amount of silver in ceramic walls and filtered water is affected by the frequency of filter use. Hence, silver levels decrease with continual filter use.

For silver nanoparticle filters, the nanoparticles may be transported out of the filter, resulting in nanoparticles in the drinking water. Ren and Smith [44] mention that nanoparticles are relatively mobile through the porous ceramic media, and the extent of mobility depends on the nanoparticle properties and water chemistry. Figure 8 shows the silver concentrations for each type of filter decreasing over the extent of the research.

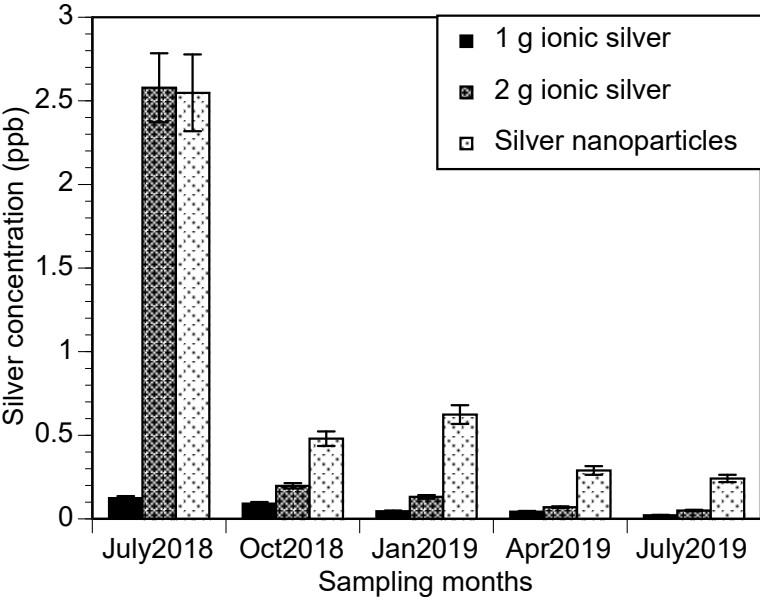

**Figure 8.** Results of experiments showing silver concentration in effluent over a 13 month period (n = 42).

Figure 8 shows that silver nitrate filters release extremely low levels of silver ions, and it has been reported in previous studies that ionic silver is not genotoxic at different concentrations [21]. Therefore, CWFs made using silver nitrate are effective in both improving microbiological drinking water quality while releasing extremely low levels of silver for water disinfection that are within the recommended EPA and WHO silver concentration levels for drinking water guidelines.

A similar study by Jackson et al. [22] performed under laboratory conditions proved that the adding of silver nitrate before firing in filter production releases low silver levels ($\leq 10$ µg/L) to the drinking water and performs consistently in microbial inactivation over time. The results indicated that silver nitrate is a viable substitute that ceramic filter production facilities could adopt. Jackson et al. [22] further attributed the surface chemistry mechanism between the ceramic and the silver nanoparticle versus the silver nanopatch to influence the differences in a silver release. The low levels of silver released by the 1 g ionic silver filters compared to 2 g in the first month could be due to the leaching of silver from the CWF into the treated water.

Brown et al. [23] performed a field study in Cambodia using a CWF painted with AgNO$_3$; their result showed good bacteria inactivation. They, however, did not report on the release of silver in the filtrate. Mittelman et al. [10] reported that CWF made with AgNPs in a laboratory-based study recorded a higher retention capacity and life span when compared to CWF made with AgNO$_3$. This complements the report of Ryner et al. [11], who found that silver release in the filtrate of AgNO$_3$ filters was significantly higher than that of AgNPs. Lower levels of AgNO$_3$ (<0.3 mg/g) yielded filtrate with Ag$^+$ that did not exceed 100 ppb, but the use of 0.3 mg/g in the CWF yielded filtrate with Ag$^+$ concentration in the range of 797 to 2697 ppb. It is noteworthy to state that the method of silver nitrate application was through the brushing method, which is different from the mode of AgNO$_3$ application in this study. Therefore, we can conclude that the method of AgNO$_3$ application can influence the levels of silver in the filtrate of CWF.

Ryner et al. [11] also reported that the level of silver released was dependent on the kind of clay materials used. The authors stated that the increase in bacteria inactivation was dependent on the concentration of AgNO$_3$ applied, with higher concentration recording better bacteria inactivation. The study of Kendarto et al. [24] reported more *E. coli* inactivation as the AgNO$_3$ concentration was increased from 0.005 M to 0.01 M, but our study found that the increase of silver ion from 1 g to 2 g did not yield significant bacteria inactivation. In a field study conducted in Limpopo Province of South Africa on the use of CWF painted with AgNPs by Hill et al. [45], the levels of silver released complied with the WHO drinking water guidelines for sampling performed on 23 households after 13 months (10.1 ± 0.46 µg/L) of filter use. However, 3% of 93 households recorded levels (112–274 µg/L) higher than the WHO guideline value after 17 months of filter use. The authors argued that the households could have left the water for a long period of time, leading to the accumulation of silver ions released from the filters.

All the households in this study reported that their main source of drinking water is municipally treated water, which is often disinfected using chlorine. The storage of the water for days creates the possibility of recontamination. A study conducted by Lyon-Marion et al. [43] reported that the use of chlorinated waters on CWF has a minimal impact on silver release from the filters; hence chlorinated water can be used as source water in CWF.

In conclusion, since the 1 g, ionic silver from silver nitrate filters release extremely low silver levels compared to 2 g and silver nanoparticles filters during water treatment, and both attain similar levels of bacteria inactivation; hence 1 g ionic silver filters are the best option to adopt. CWFs made with silver nitrate do not only make water safe for human consumption (microbiologically and in terms of silver release) but also reduce the risks of occupational exposure (to silver) to workers involved in filter making. Silver nitrate is mixed with clay, sawdust, grog and water during the manufacturing process, while AgNPs are painted to the filters after firing. Using silver nitrate, therefore, eliminates the possibility of inhalation exposure to the filter manufacturing workers.

### 3.6. Economics of the Process

3.6.1. Cost Analysis

A cost analysis of the economic benefit of substituting AgNPs with silver nitrate in the production of CWFs was carried out. Overall, the silver nitrate chemical costs less than silver nanoparticles in terms of the purchasing price per kilogram and considering that silver nitrate is locally available in South Africa, no shipping costs are incurred (2019 pricing: 1 South African Rand approximately equals 0.066 United States Dollars). AgNPs were purchased and shipped from Spain, implying that there was additional shipping cost associated with the purchase of AgNPs, as shown in Table 4.

**Table 4.** Summary of costs related to the purchase of silver nitrate and AgNPs.

|  | **Silver Nitrate** | **AgNPs** |
|---|---|---|
| **Price per kg** | R9,085 | R28,062 |
| **Shipping Cost** | R0 | R4,502 |
| **TOTAL** | R9,085 | R32,564 |

Besides the above-mentioned purchasing costs, using silver nitrate eliminates one stage labor costs as the painting step is removed because silver nitrate is added during the manufacturing process. Ryner et al. [11] reported that the use of $AgNO_3$ for CWF has an impact on its cost as it is generally cheaper compared to the use of AgNPs. This is also attributed to the absence of AgNPs in most developing countries that need the CWF, and the shipping cost usually increases the overall cost of the filters. The findings of this study also support that the use of 1 g $Ag^+$ from $AgNO_3$ reduces the price of the cost CWF compared to that of AgNPs. The use of silver nitrate during the manufacturing process also eliminates health risks for workers as research suggests that inhalation of AgNPs may result in genotoxic effects [20,21]. Elimination of these health risks will therefore avoid occurrences such as downtime at the workplace, lack of productivity and compensation claims by employees.

However, one drawback, as noted by Jackson et al. [22], is that silver nitrate filter production requires the application of silver before quality tests (pressure and flow rate tests) and if a filter fails to pass one or both tests, the silver will be wasted.

3.6.2. Unit Cost Analysis

During the manufacturing of different types of filters under study (1 g $Ag^+$, 2 g $Ag^+$ and AgNPs), the quantity of all other inputs essential in filter making remains constant, i.e., clay, sawdust, grog and water. Only the type, quantity and application method of chemical differs. Therefore, unit cost analysis is carried out to determine how the quantity of filters produced affects the total costs. Figure 9 shows the costs of producing 1000 filters using different chemicals. The unit cost analysis, therefore, concludes that it is more economical to manufacture CWFs with 1 g ionic silver from silver nitrate as they are cheaper to produce than 2 g ionic silver from silver nitrate and silver nanoparticles filters. Because of this lower production cost, filter manufacturing facilities may consider reducing the current selling price of CWFs, thus improving affordability to poor communities that lack clean water supplies.

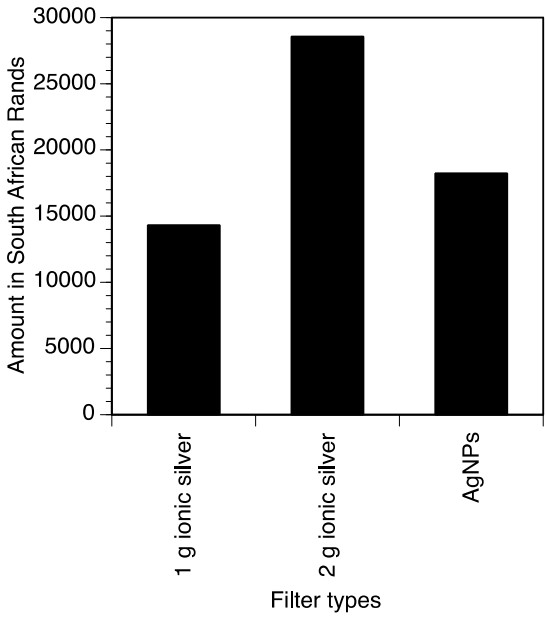

**Figure 9.** Costs associated with producing 1000 filters.

## 4. Conclusions

Silver nitrate impregnated CWF slightly performs slightly better in removing microorganisms from drinking water compared to the conventional AgNPs CWF. Therefore, silver nitrate impregnated CWFs can be adopted in the provision of safe drinking water at the household level. Both silver nitrate and silver nanoparticles CWF release silver concentrations that are below the recommended drinking water guideline (100 ppb) for silver levels. In terms of silver released, the 1 g ionic silver filter recorded the lowest levels with excellent bacteria inactivation.

This method reduces the risk of inhalation of the chemical by workers, thereby improving their occupational health and safety. One gram ionic silver from silver nitrate impregnated CWFs is a viable option to adopt because they are cheaper to produce. Silver nitrate can be purchased locally in South Africa; hence there are no importing costs associated with its use. It is therefore economical to substitute AgNPs with silver nitrate in the production of CWFs.

In summary, this method of silver application could potentially improve performance, reduce production costs, and increase the safety of production for workers as well as consumers drinking ceramic filtered water.

**Author Contributions:** Conceived and designed the experiments: N.N., J.N.E., J.A.S. and J.O.O.; performed the experiments: N.N.; contributed reagents/materials/analysis tools: N.N., J.N.E. and J.A.S.; analyzed the data: N.N. and J.N.E.; wrote the paper: N.N. and J.N.E.; participated in the editing of the manuscript: J.N.E., J.A.S. and J.O.O. All authors have read and agreed to the published version of the manuscript.

**Funding:** This project was partially funded by the University of Virginia's Jefferson Public Fellows (JPC) program. The content is solely the responsibility of the authors and does not represent the official views of the funders.

**Institutional Review Board Statement:** The study was conducted according to the guidelines of the Declaration of Helsinki and approved by the Ethics Committee of the University of Venda (protocol code: SES/18/HWR/07/2905 and date of approval: 29/05/2018).

**Informed Consent Statement:** Informed consent was obtained from all subjects involved in the study.

**Data Availability Statement:** The main data obtained in this work is contained in the manuscript. Others can be made available upon request from the corresponding author.

**Acknowledgments:** The authors thank workers from the PureMadi Dertig Filter Facility for their assistance in making filters, as well as A. Estrella from the University of Virginia, who did much of the analysis for silver levels in the effluent. The authors also acknowledge the work of T. Beddow, E. Taylor-Fishwick and M. Sutton from UVA, who were involved in survey data collection, sample collection and microbiological analysis at the beginning of the study.

**Conflicts of Interest:** The authors declare no conflict of interest.

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
