# Peer review of "Field Investigation and Economic Benefit of a Novel Method of Silver Application to Ceramic Water Filters for Point-Of-Use Water Treatment in Low-Income Settings"

_water, doi:10.3390/w13030285_

Round 1

Reviewer 1 Report

I am not qualified to comment upon techniques for detection of micro organisms

Author Response

Response to Reviewer’s 1 comment

We feel this reviewer didn’t have any comment that require action from us as he/she seems satisfied with the work.

Reviewer 2 Report

This manuscript describes a field evaluation of ceramic water filters which was undertaken to understand filter performance when a new silver application method was used in the manufacturing process. This is a helpful paper with some useful results, but there are some issues with the methodology and analysis which need to be addressed in order to draw the conclusions the authors have suggested. There are also a number of minor comments that should also be addressed in order for the paper to read more clearly and completely. As it is, many of the details are inconsistent or not described fully, which leaves the reader questioning the methods.

MAJOR COMMENTS

  1. This paper describes the use of silver nitrate in ceramic water filter manufacturing as “novel” or “new”. But in fact, in the Current Practices in Manufacturing of Ceramic Pot Filters for Water Treatment (Rayner, 2009) – this method was described as being used by various factories at that time. While it may not be the most common silver application method, I don’t think it is fair to describe it as “novel”. This should be adjusted throughout.
  2. There is some issue with the study design that needs to be addressed. The fact that some households received two filters and others received one filter would have implications on the longitudinal comparison of the filters. Presumably households with one filter would be using it twice as much as households with two filters. If a household had one filter, testing that over 3 months of normal use would be like comparing to a filter tested over 6 months (if the household had two filters). As a result of this, I am not sure if the comparison presented here is as strong as it appears. This is a severe limitation that needs to be discussed or adjusted for if possible. For example, was any survey data collected at the follow-ups to understand the frequency of filter use, or if the filters were in use at all?
  3. There is also some issue with the analysis of the microbiological data. Specifically:
    1. Typically, geometric means are reported for bacterial concentrations instead of arithmetic means. This is because of the wide variability in measured bacteria concentrations (which is also why a log scale is often used to plot results). Since the log reduction values are reported here and compared to other studies that did similar testing, I suggest re-calculating your averages and percent reduction values using the geometric mean.
    2. According to Figure 3, from January – July 2019 there was essentially no E. coli in the source water. Using these results to declare the filter performance isn’t really an accurate representation of the filter performance or its capability. This should be discussed, and/or results should be parsed out for months when the E. coli influent concentrations were higher.
    3. A statistical test needs to be used to compare the log reductions of the different groupings of filters if the aim is to show how they differ from each other. This study used very small sample sizes (10 filters with AgNP), so I imagine that there is actually no statistically significant difference between the field performance of the AgNP filters versus the AgNO3 filters. Therefore, the conclusion that “Silver nitrate impregnated CWF performs better in removing microorganisms from drinking water compared to the conventional AgNPs impregnated CWF” would not be accurate. But this needs to be tested and shown here in order to draw that conclusion.
  4. This paper would benefit from further discussion of the source water sources used by households over time, and whether that had any relationship with the filter microbiological performance or measured silver elution. It is known that water quality parameters vary with source type, but this is not shown here at all, and the source types in all cases are unknown. For example, in figure 3, the E. coli concentrations of raw water varied substantially, but it is unknown if this varied because the samples were from different source types. I recommend the following revisions:
    1. Describe if the baseline physical-chemical parameters are all from the piped water systems, and in what month they were taken (and if this was wet or dry). Also shorten the discussion of these physical-chemical results, but make it more clear the relevance they have to the remaining raw water samples collected throughout the study.
    2. Include a table to indicate what percentage of samples in each month were from what source types.
    3. Consider presenting the E. coli results sorted by source type.
    4. Consider presenting the E. coil and total coliform reductions for each month over time, rather than all combined together. And/or – consider only reporting the results when the influent E. coli started out greater than a minimum value (e.g. 10 CFU/100mL)
    5. Consider presenting the effluent silver concentrations measured from different water sources
  5. A careful review and revision of the methods needs to be conducted, to accurately describe what was done. For example:
    1. The physical/chemical test methods listed in the methods don’t match what is presented in the results.
    2. The silver spot-check results listed in the methods are not included in the results.
    3. The water sampling procedure is not at all clear. The methods should include how many times filters were tested, at what time interval, how long were they in use by the households before testing, if samples were taken from both influent and effluent water, how those samples were collected (e.g. directly from the filter outlet, or storage container, etc). None of this detail is provided.
    4. The cost analysis as presented in the methods and as presented in the results also do not match up.
  6. Throughout the results, many comparison studies are referenced. These should be introduced with a better literature review in the introduction of other field evaluations of CWFs – with microbiological performance and silver elution discussed, to set the stage for the results.

MINOR COMMENTS

INTRODUCTION

General comment: It would be helpful to describe (very briefly) how ceramic water filters are manufactured, as well as the different options for how silver may be applied (e.g. brushing, dipping, or firing in). Additionally, the purpose of applying silver should be included. That is, that silver is used for additional disinfection as well as preventing the growth of a biofilm on the filter surface.

References 1, 2, and 3 are fairly outdated (2008, 2011, 2008, respectively). There are much more up-to date references on water-borne disease rates and estimated attribution from WASH to diarrheal disease. For example, see Pruss-Ustun 2019 (https://www.ncbi.nlm.nih.gov/pmc/articles/PMC6593152/) – and the 2019 update to the WHO “Safe Water, Better Health” publication (https://apps.who.int/iris/handle/10665/329905)

[Lines 49-52] – It doesn’t seem necessary that a HWTS technology needs to be produced locally. That is one benefit to something like a ceramic filter, but there are also other filters and other HWTS options that are imported, and are also still effective.

[Line 53] Suggest changing “third world countries” to “low income countries” (similar to the article’s title) or “underdeveloped” as was used in Lines 45 and 47. Either way, these should all be consistent throughtout.

[Line 53] Define “POU” at the first use of the term

[Line 56] Define “CWF” at the first use of the term

 [Line 71] “Silver nanoparticles are painted to these CWFs to act as a disinfectant towards pathogens” – is application of silver just for this method? Or also to prevent a biofilm from forming on the surface of the filter?  

[Line 78] The WHO does not set Standards, but instead they set Guidelines. And according to the 4th edition of the GDWQ, there is no health-based guideline value for silver in drinking water, but it implies that a level of 100 ppb should be safe against health effects (https://www.who.int/water_sanitation_health/water-quality/guidelines/chemicals/silver-fs-new.pdf?ua=1)  This should be made more clear, and Reference 12 should be the WHO Guidelines or other similar reference.

[Line 89—90] “Also, it was proven that filters made with silver nitrate improved silver retention in the filter and increased the general lifespan of the filter”. This seems to be at odds with what was found in Mittelman et al (2015) – where silver nanoparticles were shown to be more likely to stay longer in the filter than silver nitrate, under a variety of influent water characteristics. This statement should maybe state that this isn’t as “proven” as it might appear.

[Line 91 – 92] “…the risk of inhalation exposure by workers manufacturing CWFs is eliminated.” If silver is still applied, but in a different part of the manufacturing process, I don’t think you can say that the risk is completely eliminated. It is just changed. Perhaps it is lessened, but there are still people interacting with the silver earlier in the process, right? This should be made more clear.

[Line 93-95] I suggest changing “effectiveness” to “microbiological effectiveness” in the study aim. You also may want to include something about confirming the silver elution for safety as well.

MATERIALS AND METHODS

[Lines 97-107] The language in the “Study Design” section is unclear and hard to follow. I suggest re-writing to be more clear. Perhaps list a chronological summary of what was done from a high level: X households were selected, a baseline survey was conducted and physical-chemical water testing was done, three types of filters were distributed to households, follow-up with each household water quality testing was done monthly for 13 months – including microbiological testing of unfiltered and filtered water samples.

[Lines 114-117] The climate data does not seem relevant to this study, so I suggest removing.

[Line 118] I suggest removing “distribution of” from the title, as the text does not discuss the coverage of these water source types. Only a list of them is included.

[Line 123] How is groundwater extracted? Is it through they unprotected wells, or protected hand pumps?

[Line 126] Please define what “grog” is. I am not familiar with this term in this context.

[Line 126 – 133] Please explain in more detail the differences in the two (or three?) filter types that were produced. For example, how was the AgNP applied? (I believe it was painted on after firing.) And how was the AgNO3 applied? (I believe it was mixed into the clay and fired in.) This needs to be explicitly stated here. Also, this section states that 50 filters were produced, but then later it states that only 30 filters were given to households. What was the reason that extras were produced and how were the 30 filters selected for distribution?

[Line 155-156] How were the households randomly selected? Were they first randomized, and then households were asked if they wanted to participate? Or was a list of households willing to participate created first, and then the selection was done from those households?

[Line 159-160] How was it determined which households would receive the second filter? Were households instructed to use both filters? Or is it possible that they only used one filter during the study period?

[Line 160] Is “impregnated” the correct term here for the silver nanoparticles that are painted on?

[Lines 161 – 168] This section does not describe the same water quality parameters that appear in the results section (e.g. color) – and include extra ones (e.g. dissolved oxygen). Fix the methods here to match the results that are presented.

[Line 184-185] I suggest removing what tools the kit included.

[Line 198] What costs and expenses were included? Simply the raw material costs? And it’s not clear what expenses would be incurred by using silver nitrate.

[Line 198-199] Please explain how “the increased revenue that would come from using silver nitrate” was determined. (Also note that this was not presented in the results)

RESULTS

[Line 220] How is the source water treated? If it is chlorinated, does that have any effect on the filter or interaction with the silver?

[Line 231-232] I am not sure I understand what is meant by this sentence: “100% of the households have gone beyond 24 hours without using their stored water and because water is a scarce commodity, they do not discard it but rather continue to drink it.” Is this saying that households still drink water that has been stored for more than 24 hours? To me, that is not surprising, since it has already been stated that water is often unavailable for several days at a time. I am not sure this sentence is needed.

[Line 233] I suggest putting the reference to Figure 2 at the end of the sentence rather than in the middle for clarity.

[Line 238] Figure 2 caption should clarify that this was a survey at baseline – before receipt of the filters.

[Line 253-274] The long description of color in the water does not add very much value to the paper. I suggest removing or shortening.

[Line 283] Consider changing the phrase “in order to signal microbial water quality changes after the use of CWFs” to “in order to test the microbiological effectiveness of the CWFs in households.”

[Figure 3] What might be an explanation for the difference in E. coli concentrations between July 2018 and July 2019? I understand changes due to seasonality, but to be so different in the same month is confusing.

[Line 289] This line states that results are “regardless of the source or storage conditions”... but those storage and source conditions are not shown here. I suggest revising or removing.

[Line 289 – 296] The long description of the meaning of E. coli versus total coliform results is a maybe unnecessary. Typically, E. coli is used as an indicator organism to indicate water safety, while total coliform may be used as an indication of filtration effectiveness (more so than water safety).

[Line 297-298] It states here that the source type at the beginning of the study was the municipal water supply. What about other times later in the study?

[Line 319 & Line 346] The description of the median values used in these two situations is not correct. The median is ONE value for your data set, and cannot be a range.

[Tables 3 and 4] The standard deviation should be listed as Average (SD), rather than Average +/- SD. The standard deviation represents variability in the measure, but shouldn’t be confused with a confidence interval, which may be described by “+/-“

[Table 4] Change the text in the table from total coliforms to E. coli.

[Figures 4 and 5] These use an arithmetic y-axis scale, as opposed to the log scale used in Figure 3. It seems that the same scale should be used across the board.

[Line 353-355] This description of training provided to the households may be better located within the methods section.

[Line 379] This study used a small sample size, so I’d use the word “prove” very cautiously

[Figure 6] I understand why you used different scales in the two figures, but would suggest sticking with the log scale for both. The E. coli graph range would probably just go from 1 – 10.

[Table 5] I don’t think it’s helpful to average the two removals of E. coli and total coliform. This doesn’t really create a meaningful metric – each should be considered on its own. I suggest removing this column from the table, and the accompanying discussion.

[Line 388-390] The difference in log removals quoted from Ref 16 between the two filter types is not really substantially different (removal of 4.1 versus 3.9 E. coli removals) and might have actually depended more on the influent water quality in the lab studies than in actual performance difference. This should be carefully reported here as a real indicator of which performs better.

[Line 403-407] This should be included in the methods, not the results section.

[Figure 7] Please indicate in the figure or in the text how many silver samples from each filter type were analyzed at the different times. It seems like a small sample size, so this should be made transparent.

[Figure 7] Is there a reasonable explanation as to why the silver elution of the 1g AgNO3 and 2g AgNO3 were so different in the first month of use? This should be discussed.

[Figure 7] Are these silver concentration values similar to what has been measured in other studies? They seem quite low, but I am not familiar with what would be expected.

[Line 461] Consider renaming “3.6.2. Sensitivity Analysis” to “unit cost analysis” or something similar. This doesn’t seem to be a sensitivity analysis, but instead just another economic model used to compare prices. Additionally, the methods used to determine these unit costs should be included in the methods section. Does it account for both materials and labor costs?

Author Response

Thanks for your comments. Kindly see the attached document for our responses to each of your valuable comments which has helped to improve the clarity of our manuscript

Reviewer 3 Report

Ndebele et al. presented an interesting study to test ceramic water filters (CWFs) with real-world use in South Africa. CWFs fabricated with silver nitrate (1g) showed a higher removal efficiency in average for total coliforms and Escherichia coli (E. coli) than CWFs impregnated with silver nanoparticles (AgNPs). The results are very relevant due to the different variables that could affect the CWFs performance in comparison with controlled variables in lab tests.

The authors need to double check the manuscript in terms of grammatical errors and provide more information especially on the methodology section.

Format/English changes required/suggested

Line 17: Add chemical formula of silver nitrate (AgNO3).

Line 19: Add the complete name of the microorganism the first time that is used in the manuscript Escherichia coli (E. coli) instead of “E. coli”.

Line 38-40: The sentence “Drinking water is a medium … waterborne diseases” is very long. Please try dividing it into two sentences.

Line 74, 76, 79 and 81: use bullet symbols or letters instead of numbers at the beginning of each item.

Line 97 and 213: it is recommended to add a transition sentence between the title and sections.

Line 103: provide the concentrations/amount of silver used in each of your reactors instead of “different concentrations of silver nitrate”.

Line 126 and line 131: information of silver nitrate reagent need to be added. For instance purity, manufacturer.

Line 133: Brief description about the method used for the fabrication of each filter should be added either in the main document or in the supplemental information section.

Line 139: Delete “map showing” of the map title. Add country to the map title.

Line 158 Change “fifteen” by “Fifteen”.

Line 166: change “(U.S. Environmental protection agency method 180.1)” by “according to U.S. Environmental protection agency method 180.1

Line 176 and 180:  Double check superscript of your numbers, “10-7” and “35oC”.

Line 181: add CFU/100 mL after “.. colony forming unit per 100 mL…”. What were the negative controls used during the total coliforms/ E. coli quantification? i.e. Phosphate buffer solution (PBS), etc.

Line 187: detection limit of the Hach kit should be added.

Line 188: details about the GFAA equipment should be provided. For instance brand, detection limit

Line 190, 328: Was the silver ion release quantified from the CWFs with AgNPS? If so, add information

Line 198: equivalence of Rands to $US can be added for comparison purposes

Line 189, 214,218, 220, 231, 368:    avoid starting your sentences with numbers: 10, 30, 100, 100, 10. It is recommended to use ten, thirty, one hundred, etc., respectively.

Line 244 Use “color” instead of “colour”.

Line 258 and 287: The authors showed the average values but the total number of samples was not provided on the table or document. Please add value.

Line 293 Change “faecal” by “fecal”

Line 308 “impregnated with silver” please specify what kind of silver was used in that study. i.e silver nitrate, silver nanoparticles, etc.

Line 316 add units to the third column of the table 3 “average total coliforms” (CFU/100 mL?). The authors mentioned in the caption of the table  3  “one-sample statistics”, the authors meant one-sample t test. Information about statistical analysis was not provided in the manuscript. Please add the respective information.

Line 320 to 324: The sentence “Ceramic filters … hygiene behavior.” is very long. Please try dividing it in two.

Line 325 and 351: units were not provided on the y axis.

Line 340: Add information about what kind of silver was used in that study. i.e silver nitrate, silver nanoparticles, etc.

Line 334, 343, 352, 381: Use italic font for species names: E. coli  instead of E. coli.

Line 358: it is the first time that the authors provided information about the ceramic filter apparatus used during the test. Please add the respective information about the arrangement, contact time in the methodology section.

Line 381: add the letter “a” and “b” to the respective graphs or mention in the caption right or left.

Line 388-39: The authors used the reference [16] to align their results with previous studies, but additional reference(s) should be added in the discussion.

Line 406 and 407: The authors mentioned “..2-month intervals were randomly selected …” but the graph showed intervals of 3 months. Please provide clarification.

Line 436; Change “they” by 1g silver nitrate

Questions

  1. Why did the authors choose silver nitrate instead of other silver salt, (for instance silver chloride) during the CWFs fabrication?
  2. Line 125-133: Production of CWFs is a very important section and more information can be provided to identify similarities and differences between filters. Please add information about hydraulic retention time or pore volume or porosity and dimensions of each filter/apparatus. Are the AgNPs used in the CWFs coated? What is the amount of silver impregnated in the CWFs with AgNPS?
  3. Figure 3 and 4 showed the differences of inflow and outflow in terms of total coliform and coli concentration. Could the authors please clarify if those results take in consideration a specific CWF or is it the median of all of the collected samples in all of the filters? Add the respective information in the manuscript.
  4. Total coliform and coli concentrations of raw water changed over time according to the figure 3. What happened after December 18 that E. coli concentrations decreased drastically in the raw water samples?
  5. The authors mentioned in the manuscript that different variables were collected including turbidity, silver concentration, etc, but they did not mention the controls used during the sample collection and sample analysis. What kind of controls (positive/negative) were used during those 12 sampling months?
  6. How were trained those families with both CWFs with silver nitrate and silver nanoparticles to use both filters at the same frequency?
  7. The removal efficiency is function of the initial concentration and the final concentration (in this case very similar for the three scenarios according to the figure 6). Was it a coincidence that all of the final concentration were around the same value or is a removal efficiency limit by the CWFs at that concentration?

Author Response

Thanks for your comments. We have responded to each of them including the questions you raised. Kindly see the attached document.

Reviewer 4 Report

Overall, the paper is well structured with clear methods.

The only suggested correction is on line 104..."silver levels in THE effluent..."

Author Response

Response to Reviewer’s 4 comments

Thanks for your kind comments on our manuscript. We have done all the required changes you suggested.

Comments

Response

Overall, the paper is well structured with clear methods.

Thanks for your great comment

The only suggested correction is on line 104..."silver levels in THE effluent..."

We have corrected this in the revised manuscript. Kindly see the track changes that were done.

Sincerely,

Edokpayi Joshua, Ph.D.

Round 2

Reviewer 2 Report

The authors have addressed most of the comments adequately; however, a few inconsistencies remain in the presentation of results that I believe should be addressed in the final manuscript.

  1. The authors’ response to my previous comment about the E. coli results varying over time as shown in Figure 3 (which was also mentioned by another reviewer) hasn’t been adequately addressed. The response was that there was an issue in the graph’s scale. The revised version includes the same graph with a new scale, but the values shown in the new graph are actually different than the values in the previous graph starting in Jan 2019. Please explain this inconsistency.
  2. For the microbiological results, the sample sizes aren’t clear to me, and seem inconsistent in many cases. Figures 3 & 4 state n=60. Is that 60 samples per month? Or 60 samples total? But then Table 4 says n=390, and Figure 4 indicates n=138. These three places I think are showing the same data in different configurations, but the sample sizes don’t match up between them.
  3. Tables 4 & 5 both refer to a t-test, but it does not appear that a t-test was actually done here. Was the idea to test if the raw water was significantly different from the filtered water? If so, then provide what the result was (p-value) and interpretation of the result. I’m not sure if a t-test is necessary to make your point in this case, but if you include it, then the test results need to be actually described. Currently what the table has is not a t-test, but just the average values and percent removal.
  4. Microbiological results are inconsistently described. In tables 4 & 5, the averages (arithmetic means) are reported, yet in the text, the medians are reported. It makes most sense to choose the measure of central tendency that fits your data best and use that throughout. For non-normal distributions like this one, if the geomean is not used (as previously commented), then the median value is probably most appropriate.
  5. Figure 7: Again I suggest including the sample size in the figure or caption.
  6. Line 581: Suggest removing the word “random”

Author Response

Thanks so much for your precious time in reviewing our manuscript. We have now reviewed the manuscript based on your valuable comments and have given explanations that were needed for some of the changes you observed in the previous version. Please kindly see the attached document.
